# Concentration of asset owners exposed to power sector stranded assets may trigger climate policy resistance

Angelika von Dulong [1,2] ✉

Thoroughly assessing the owners and distribution of stranded assets in a 2 °C scenario is essential to anticipate climate policy resistance. We employ novel data to analyze owners and incidence of asset stranding in the power sector globally. We show that Asia-Pacific, Europe, and the US are highly exposed to stranded assets, especially coal plants. Stranded assets are highly concentrated in a few asset owners in some countries (e.g., India). Even if owners are more equally exposed (e.g., in the US) they can vary considerably in the asset stranding timing due to differences in plant fleets' age profile. European, US, and Chinese asset owners own large shares of stranded coal plants abroad. Listed owners may face stranded assets of up to 78% of their share price or more than 80% of their equity. Asset stranding exposure positively correlates with ownership of alternative energy assets. India stands out owning many stranded assets but little alternative energy.

Reaching the 2 °C climate goal requires the implementation of stringent policies to transform the energy sector. This includes leaving fossil fuels unextracted[1–3] and prematurely retiring fossil fuel-burning energy infrastructure[4], also referred to as "asset stranding"[5]. The success of such policies potentially hinges upon their interaction with stranded assets[6]. Fierce opposition to policies has been shown to be formed by adversely affected asset owners[7,8]. Accounting for such resistance is crucial for producing realistic policy advice and for proposing feasible policies[9,10].

To assess potential sources of resistance to climate policies and to gain a better understanding of who has high stakes in national policy formation and international climate negotiations we ask: Who are the owners of power sector stranded assets across the globe and how are stranded assets distributed between them? Further, resistance to climate policies may be moderated if affected asset owners are also invested in alternative energy assets—potentially even benefiting from these policies. Thus, we ask whether asset owners' ownership of alternative energy assets correlates with asset stranding exposure.

The extant literature on power sector asset stranding focuses mostly on the global or country level[11], while this paper primarily targets the asset owner level. To reach the 2 °C Paris goal, coal power plants must retire decades earlier than historically[12,13]. Put differently,

globally, only 42–49% of (operating and pipeline) power plant generators can be utilized until the end of their economic lifetime[14], and 300 GW of coal-fired capacity commissioned between 2011 and 2014 must be stranded to reach the 2 °C climate change target[15]. Depending on the policy stringency and the time horizon global stranded assets in coal capacity range between US$150 billion and US$1.4 trillion[16,17]. While these papers are important contributions to understanding the extent and associated costs of asset stranding in the power sector, they do not reveal information on affected stakeholders below the country level, especially how costs are distributed over the direct owners of assets and owners higher up the ownership tree. This, however, is crucial for anticipating resistance to policies and providing realistic policy recommendations.

To the best of our knowledge, only a few papers analyze asset stranding at a more fine-grained level. In the power sector, Breitenstein et al.[18] quantify stranded assets of German power companies due to the country's coal phase-out. They show that individual companies may suffer absolute losses worth €4.8 billion and more than €7 per share outstanding if the coal phase-out is implemented in 2030 as opposed to 2038. The asset owner-level exposure to asset stranding has further been studied outside the power sector. For instance, regarding the upstream fossil fuel producing sector, Semieniuk et al.[19] trace global

¹Humboldt-Universität zu Berlin, Berlin, Germany. ²Berlin School of Economics, Berlin, Germany. ✉e-mail: angelika.von.dulong@hu-berlin.de

stranded assets from the oil and gas sector to the ultimate owner and find that predominantly non-listed investors headquartered in countries of the Organisation for Economic Co-operation and Development (OECD) are exposed to stranded assets. Although these studies provide key findings for asset stranding at the asset owner level, they focus either on German companies solely or on the upstream fossil fuel producing sector. This paper aims at filling this literature gap by assessing power sector asset stranding at the asset owner level globally.

In this study, stranded assets are computed using a unique combination of two data sets. The first data set from Asset Resolution covers assets around the globe linked to their direct owner and the entire ownership tree of asset owners owning the direct owner[20]. We use a subset of the data, which focuses on the power sector and includes, among others, information on power plant operating capacity, age, location, and ownership structure. We match this data set with data from the International Energy Agency's (IEA) World Energy Outlook 2021[21]. The IEA data provides a scenario on regional fossil fuel power capacities, which allow for a sustainable development in line with the 2 °C goal ("Sustainable Development Scenario"). If this climate-compatible capacity is exceeded by the operating capacity as given by the first data set, we successively identify power plants as stranded (oldest plants first in the baseline analysis) until the climate-compatible and operating capacity are in line. Then, we compute stranded assets as power plants' overnight capital costs (OCC, for a definition see the Supplementary Information), which are not recovered due to premature decommissioning. Finally, we aggregate these stranded assets at the asset owner level.

Our results suggest that prematurely decommissioned power plants are predominantly located in Asia-Pacific countries, Europe, and the US, and they mostly use coal as energy input. Climate pledges announced by countries in Asia-Pacific and Europe (outside the European Union) largely fall short of those required for a sustainable development. Thus, compared to asset stranding in line with announced pledges a considerable amount of additional assets—especially coal power plants—must be stranded to reach a sustainable development in those countries. We show that the distribution of stranded assets across asset owners varies strongly between countries. For instance, in India one single asset owner owns the majority of stranded assets, which is in stark contrast to the US, where stranded assets are much more equally distributed across asset owners. Zooming in on the US, we find that even if asset owners are equally exposed to stranded assets, they can differ considerably in the timing of asset stranding due to differences in the age profile of power plant fleets. Often, the location of stranded power plants is quite different from the location of asset owners ultimately owning these plants. For

instance, European, US, and Chinese asset owners own a large share of stranded coal power plants located in foreign countries. Asset owners listed on stock markets may find assets worth up to 78% of their share price or more than 80% of their total equity stranded. Listed asset owners in OECD countries are more able to buffer their exposure to stranded assets with their equity compared to non-OECD asset owners. Finally, there is a positive correlation (Spearman's $r = 0.69$) between asset owners' ownership of alternative energy assets and asset stranding exposure. Across regions, China and India stand out: Both are highly exposed to asset stranding but compared to China, India shows relatively little ownership of alternative energy assets.

## Results

### Stranded assets across regions and fuels

In total, almost 2.8 TW of fossil power plant capacity must be stranded globally between 2021 and 2050 to be in line with the IEA's Sustainable Development Scenario. Employing our method of assessing the monetary losses of prematurely decommissioning these plants, this translates into stranded assets worth more than US$ 500 billion (for sensitivity analyses see the Supplementary Information). Figure 1 presents stranded assets across regions and fuels. Regions most affected are Asia-Pacific, Europe, and the US (for the plant-level spatial distribution of stranded assets see the Supplementary Information). Predominantly power plants using coal as energy input are stranded. Thus, these regions may face social repercussions and policy resistance, in particular resulting from the implementation of coal power plant shut-downs—this, however, requires further research.

Opposition to climate policies fostering a sustainable development may be particularly strong if such policies result in more stranded assets than those under currently announced policies. To quantify stranded assets in line with policies currently announced, we employ the IEA's Announced Pledges Scenario, which assumes implementation of all recently announced 2030 climate targets and longer term net-zero pledges (for scenario details see the "Methods" section). We define the stranded assets ambition gap as the difference between stranded assets in the Sustainable Development Scenario and those in the Announced Pledges Scenario. Quantifying asset stranding due to current announced climate pledges results in stranded assets worth US $212 billion. Thus, there is an ambition gap between the two scenarios equivalent to stranded assets worth almost US$300 billion. Figure 2 shows the distribution of this stranded assets ambition gap across regions and fossil fuels. Asia-Pacific countries excluding China, India, and Japan alone make up around US$100 billion of the stranded assets ambition gap. Announced pledges from India, China, and European countries (outside the European Union) largely fall short of targets required for a sustainable development resulting in an ambition gap of

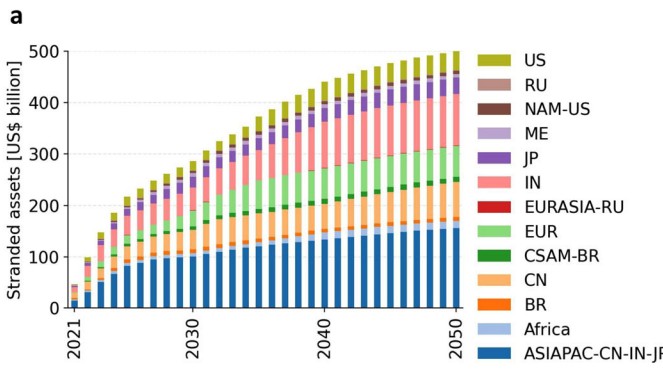

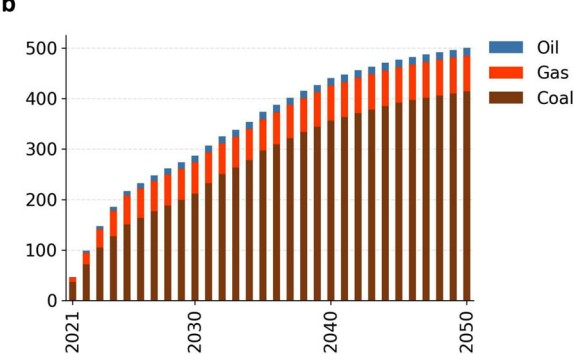

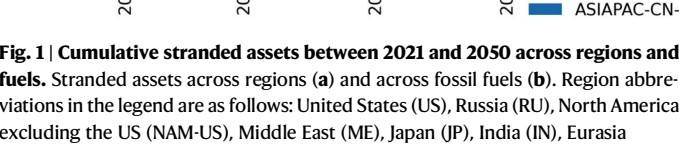

**Fig. 1 | Cumulative stranded assets between 2021 and 2050 across regions and fuels.** Stranded assets across regions (**a**) and across fossil fuels (**b**). Region abbreviations in the legend are as follows: United States (US), Russia (RU), North America excluding the US (NAM-US), Middle East (ME), Japan (JP), India (IN), Eurasia excluding Russia (EURASIA-RU), Europe (EUR), Central and South America excluding Brazil (CSAM-BR), China (CN), Brazil (BR), and Asia-Pacific excluding China, India, and Japan (ASIAPAC-CN-IN-JP).

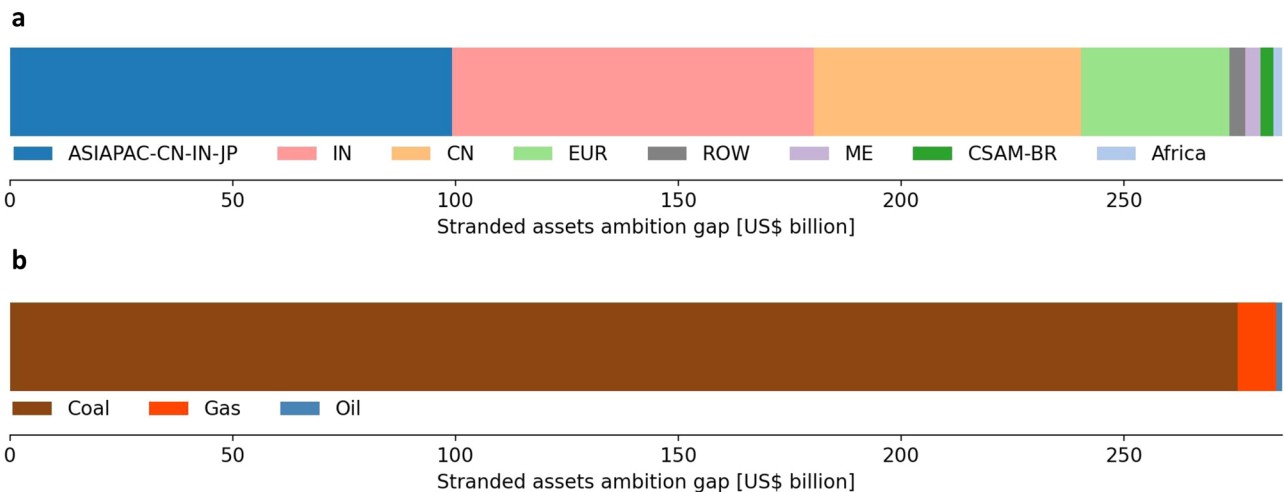

**Fig. 2 | Stranded assets ambition gap.** Stranded assets ambition gap across regions (**a**) and across fossil fuels (**b**). Region abbreviations in the legend are as described in Fig. 1.

stranded assets worth around US$80, 60, and 33 billion, respectively. Across fossil fuels, the ambition gap is largely driven by insufficiently stringent pledges targeting the phase-out of coal power plants. Globally, coal power plants worth more than US$275 billion would need to be stranded in addition to pledged shut-downs to be in line a sustainable development.

**Distribution of stranded assets across direct and parent owners**

At the country level, opposition to climate policies may be shaped by the distribution of stranded assets across owners. For instance, if stranded assets are concentrated in a few asset owners coordination of resistance may be easier compared to a situation, where many owners are relatively equally exposed to asset stranding. Using the ownership information in the Asset Resolution data (for details see the "Methods" section), we aggregate stranded assets for each power plant at the direct owner level. Figure 3 shows the top 10 direct owners most exposed to stranded assets in selected countries. The distribution of stranded assets across direct owners varies greatly between countries. In both China and India, a single direct owner is heavily exposed to asset stranding, while other direct owners headquartered in these countries suffer much less. In contrast, in the US, stranded assets are much more equally distributed between direct owners. Japan presents a mixed case, with two direct owners being more exposed to asset stranding than the rest.

Direct owners may be (partially) owned by parent owners, who are at the top of an ownership tree (for details see the "Methods" section). These parent owners can be invested in a magnitude of direct owners amplifying their stranded assets exposure. If highly exposed parent owners are nation states, resistance to policies may then not only be formed at the country level but instead shape international climate negotiations. Figure 4 contrasts the distribution of stranded assets across direct and parent owners globally. Apart from one outlier, namely NTPC Limited headquartered in India, stranded assets are relatively equally distributed across direct owners at the global level. Aggregated at the parent owner level, a great share of stranded assets is concentrated in four owners, all of which are nation states: The People's Republic of China, the Republic of India, the Republic of Korea, and the Republic of Indonesia, respectively. These parent owners may have high stakes at international negotiations on climate policies.

Parent owners can be invested in direct owners in various countries, exposing them to energy transitions globally. Then, opposition to policies may not be limited to the domestic country. Figure 5 depicts the difference in total stranded assets faced by parent owners

headquartered in a region and total stranded assets from prematurely retired power plants in the same region. Panel (a) shows that parent owners in Europe, the US, China, Japan, and the Middle East own coal power plant stranded assets located in regions outside their headquarters. On the flip-side, regions such as Asia-Pacific excluding China, India, and Japan do not own coal power plant stranded assets worth more than US$10 billion located in this region. Panel (b) aggregates stranded assets of all fossil fuels and demonstrates that North America excluding the US and the Middle East are highly exposed to stranded assets from other regions, while more than 40% of stranded assets located in Central and South America excluding Brazil are owned by foreign parent owners.

Finally, the distribution of stranded assets across time can be crucial to anticipate policy resistance. As an example, Fig. 6 shows the gas power plant capacities of two US headquartered parent owners, Vistra Corporation and Duke Energy Corporation, between 2020 and 2050. They are similarly exposed to gas stranded assets resulting from climate targets set for the US: Vistra Corporation and Duke Energy Corporation face losses as high as US$1.6 and US$1.5 billion, respectively. However, their distribution of stranded assets varies across time as their gas power plant fleets differ in age profile. Vistra Corporation in Panel (a) finds most of its gas capacity stranded between 2035 and 2045. A major share of Vistra Corporation's stranded capacity has more than 85% of OCC recovered by the time of stranding. This is in stark contrast to the stranded capacity of Duke Energy Corporation in Panel (b). Almost half of its capacity is still operating or retired with fully recovered OCC by 2050. Major stranded assets occur around 2025 when capacities with less than 75% of OCC recovered are stranded. This example demonstrates that parent owners with similar initial fossil capacities headquartered in the same country may differ considerably in the timing of their asset stranding exposure due to the age structure of their respective power plant fleet.

**Stranded assets owned by listed parent owners**

As depicted in Fig. 3, listed asset owners in various countries are among the top most exposed entities. These owners may oppose climate policies if resulting stranded assets depress their share prices. Panels (a) and (b) in Fig. 7 show stranded assets of listed parent owners by means of stranded assets per share outstanding and as a percentage in share price, respectively. Listed parent owners may suffer from stranded assets as high as US$24 per share outstanding or up to 78% of their share price. On average, listed asset owners in OECD countries are more exposed to stranded assets according to both measures. While two-thirds of stranded assets are owned by parent owners

**Fig. 3 | Stranded assets of the top 10 most exposed non-listed (blue) and listed (red) direct owners headquartered in the respective countries.** Each bar represents total stranded assets of a direct owner, which are located in the same country as the direct owner's headquarter, namely in China (**a**), India (**b**), the US (**c**), and Japan (**d**). For names of the direct owners represented by capital letters for conciseness, please refer to the Supplementary Information.

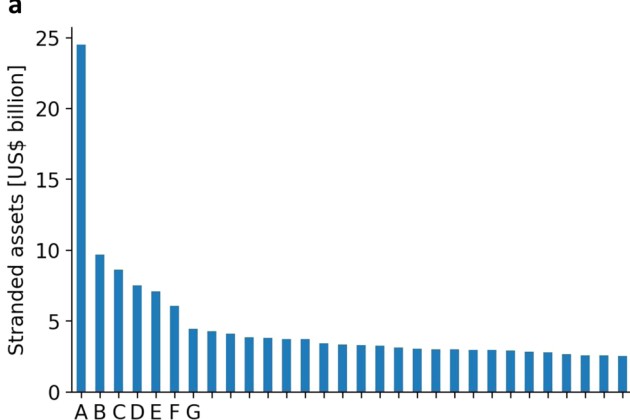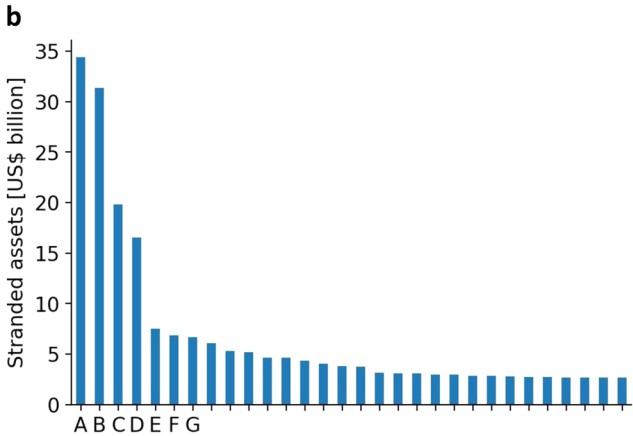

**Fig. 4 | Stranded assets distribution of direct and parent owners globally. a** As in Fig. 3, each bar represents total stranded assets of a direct owner, which are located in the same country as the direct owner's headquarter. **b** Each bar represents total stranded assets of a parent owner, which may not be located in the same country as the parent owner's headquarter. For names of the direct and parent owners represented by capital letters for conciseness, please refer to the Supplementary Information.

headquartered in non-OECD countries, this pictures shifts for parent owners listed on stock exchanges. In total, listed parent owners headquartered in OECD countries own stranded assets worth US$124 billion as opposed to non-OECD headquartered listed parent owners with stranded assets worth US$40 billion. Thus, even if parent owners in non-OECD countries show stronger exposure to stranded assets, shareholders may find stranded assets owned by listed parent owners

headquartered in OECD countries more concerning (for additional results on shareholder engagement see the Supplementary Information).

Asset owners may also resist climate policies if they are unable to cushion stranded assets with their equity. Panel (c) in Fig. 7 shows listed parent owners' ratio of stranded assets to total equity. Stranded assets may eat up more than 400% of listed parent owners'

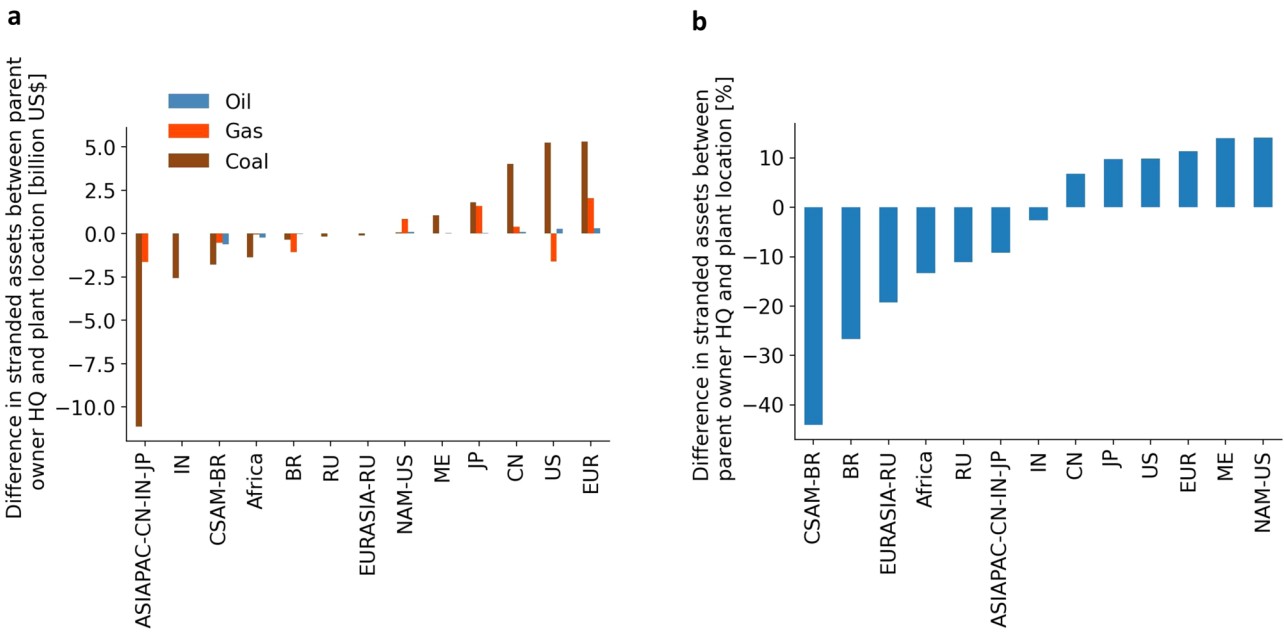

**Fig. 5 | Difference in total stranded assets faced by parent owners headquartered in a region and total stranded assets from power plants located in the same region. a** Difference in stranded assets between parent owner headquarter and plant location differentiating between fossil fuels. **b** Difference in stranded assets between parent owner headquarter and plant location aggregated over fossil fuels. Region abbreviations on the horizontal axis are as described in Fig. 1.

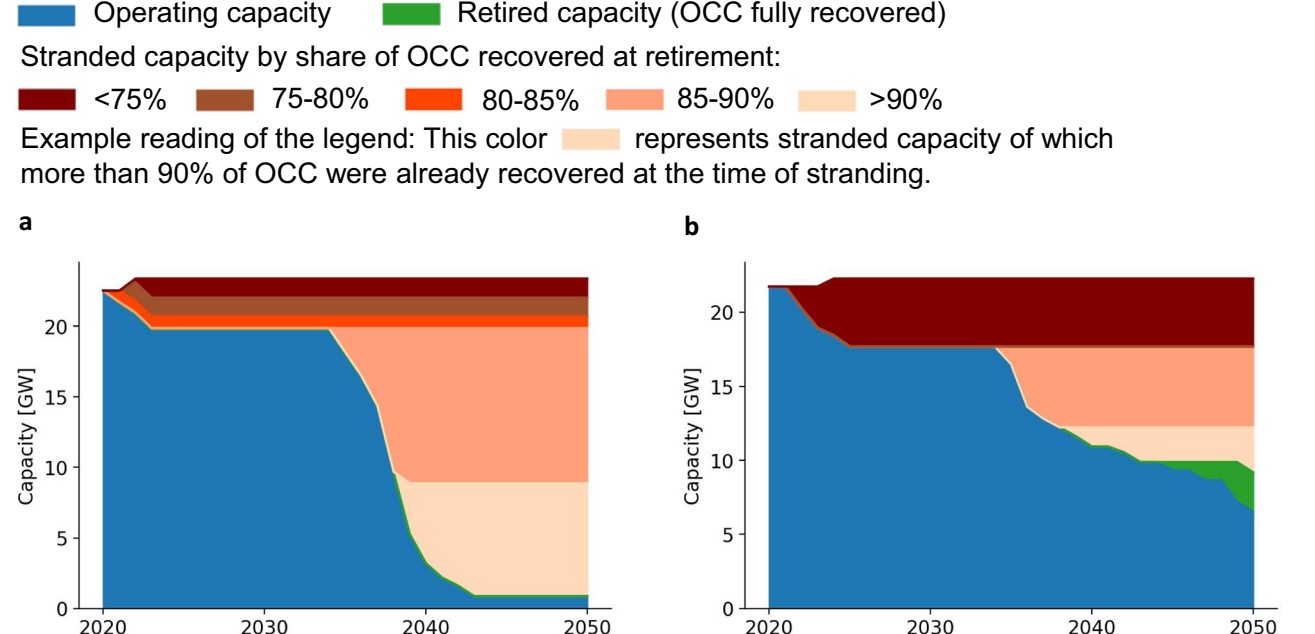

**Fig. 6 | Gas power plant capacities of two US headquartered parent owners between 2020 and 2050.** Gas power plant capacities of Vistra Corporation (**a**) and Duke Energy Corporation (**b**).

equity (or 80% for those owners where stranded assets do not exceed equity). On average, listed parent owners headquartered in non-OECD countries show higher levels of stranded assets to equity. Thus, they are less able to buffer their stranded assets exposure with the equity they own. This result is driven by the difference in total equity, which parent owners own on average: While listed parent owners in OECD and non-OECD countries are, on average, about equally exposed to stranded assets, those headquartered in OECD countries show higher levels of total equity (see below in the Supplementary Information).

### Stranded assets and alternative energy assets

While the energy transition leaves fossil power plants stranded, it also requires a massive ramp up of alternative energy assets, i.e., renewable and nuclear energy power plants. Investments in alternative energy assets may help regions and asset owners balance or mitigate their exposure to stranded assets. This could in turn reduce resistance to climate policies. Figure 8 shows ownership of stranded assets and alternative energy capacity by regions and parent owners across the globe. The regions in Panel (a) of Fig. 8 can be roughly summarized as three clusters. The first cluster consists of regions that are moderately

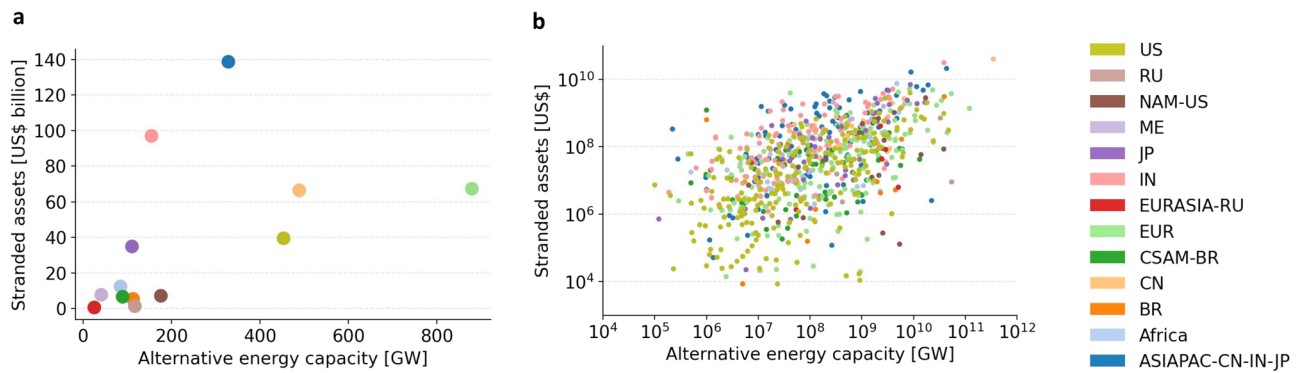

**Fig. 7 | Stranded assets of listed parent owners headquartered in OECD and non-OECD countries.** Listed parent owners' stranded assets per share outstanding (**a**), as a percentage in share price (**b**), and as a ratio to total equity (**c**). In each boxplot the box extends from the first to the third data quartile. In the box, the median of data is represented by a vertical line. Whiskers extent the box by a factor of 1.5 of the inter-quartile range and flier points exceed those whiskers. Estimations of stranded assets per share outstanding and as a percentage in share price in (**a**)

and (**b**) depend on an asset owner's debt ratio. Debt ratios differ considerably across industries, countries, and time[30]. Given that our sample of asset owners spans many industries and countries, we assume a debt ratio of 0.6. Higher debt ratios would decrease the estimations by the exact same proportion. Negative values in (**c**) arise if listed parent owners have negative total equity on their balance sheet.

**Fig. 8 | Stranded assets and alternative energy capacity owned by regions and parent owners globally. a** Stranded assets and alternative energy capacity across

regions. **b** Stranded assets and alternative energy capacity across parent owners. Region abbreviations in the legend are as described in Fig. 1.

exposed to stranded assets and own large shares of alternative energy capacity. It includes the US, China, and Europe. A second cluster is made up of India and Asia-Pacific excluding China, India, and Japan, characterized by high levels of stranded assets exposure and moderate to

low degrees of ownership over alternative energy capacity. The third cluster includes all remaining regions owning relatively little stranded assets and alternative energy capacity. At the regional level, the first cluster may be best suited to balance exposure to stranded assets with

alternative energy assets. In contrast, India and Asia-Pacific excluding China, India, and Japan may strongly resist the implementation of climate policies or the announcement of stringent climate pledges—an avenue for future research with high policy relevance.

Panel (b) in Fig. 8 disaggregates regional stranded and alternative energy assets and depicts parent owners. The plot demonstrates that parent owners differ considerably in their exposure to stranded assets and ownership of alternative energy assets. The People's Republic of China stands out being highly exposed to stranded assets and owning large alternative energy capacities (note the logarithmic scales in the axes of this panel). The majority of parent owners is either exposed to asset stranding with no ownership of alternative energy assets or vice versa. These parent owners, however, are not displayed due to the logarithmic scales. Focusing on parent owners exposed to stranded assets, there is a positive correlation (Spearman's $r = 0.69$) between their ownership of alternative energy assets and asset stranding exposure.

## Discussion

There are some limitations to this analysis. First, we only assess climate policy-induced stranded assets. Assets may also strand due to climate impacts or transition risks (cf. ref. 22), which are not directly linked to climate policies such as changing social preferences (cf. ref. 23). Second, the IEA scenario assumptions on climate policies, prices, technological progress, behavior, etc. are crucial for the valuation of stranded assets. For instance, the stringency and the design of policies can affect the distribution and absolute value of stranded assets. Energy efficiency improvements at the plant level, advances in carbon capture, utilization, and storage (CCUS) retrofitability and net-negative emission technologies could reduce stranded assets. On the flip-side, changes in preferences and the diffusion of low-carbon technologies may increase stranded assets (cf. ref. 24). Further, if the IEA employed plant-level capacities rather than aggregate regional capacities this could alter the IEA model's output capacities in the various scenarios. This would be in turn affect our stranded assets estimates. Third, this study assesses stranded assets as sunk costs. Stranded assets in terms of lost profits may be a lot higher with important implications for the feasibility of climate policies.

This study focuses on asset stranding at the asset owner level in the power sector but the Asset Resolution data likewise allow analyzing other sectors, which lack research despite their exposure to transition risks[11]. For instance, future studies could adapt the methodology employed in this analysis to assess asset owners' exposure to stranded assets in the transport sector (automobile, shipping, and aviation) and the industry sector (steel and cement). Such insights are highly relevant for policymakers: By analyzing the extant and distribution of power sector stranded assets at the asset owner level, the results in this paper can support anticipation of opposition to climate policies including lobbying efforts. This is crucial for the successful implementation of climate policies.

## Methods
### Methodology

In our baseline analysis, we identify fossil fuel power plants across the globe that must be stranded between 2021 and 2050 to achieve the 2 °C Paris goal. This requires comparing the capacity, which is consistent with the 2 °C Paris goal, with the operating capacity in a given year and region, i.e., a country or group of countries. If the climate-compatible capacity is exceeded by the operating capacity, we identify the oldest power plant as "stranded" and deduct its capacity from the operating capacity (cf. ref. 17). We repeat this step successively until the operating capacity is in line with the climate-compatible capacity in a given year and region. In a sensitivity analysis we strand power plants located in rich (poor) countries first. For simplicity we assume that power plants are always stranded entirely, so that a plant's capacity

cannot be partially reduced. The operating capacity in the analysis' base year of 2020 is composed of all active power plants in 2020 and it includes pipeline capacities for the years thereafter. In an additional analysis, we identify power plants to be stranded if all major national climate pledges are to be fulfilled. Since these pledges are not sufficient to achieve the 2 °C Paris goal[25], this analysis facilitates assessing an ambition gap in terms of stranded assets.

Once we have identified power plants to be stranded, we compute the monetary losses that accrue to the asset owners of these power plants. There are different definitions and measures of stranded assets' monetary value in the literature. One option is to compute lost profits, i.e., the difference in profits between scenarios (cf. refs. 18,19). This measure covers many aspects of stranded assets' monetary value relevant to their owners. At the same time, however, it requires a number of assumptions including the development of prices, demand, and behavior, input substitution between energy sources, as well as changes in costs, policies, and technological progress. Further, lost profits are highly dependent on the policy design[2]. For a global analysis, individual assumptions for every power plant, country or region may be necessary for this measure. Another option is to solely consider unrecovered overnight capital costs (OCC) associated with stranding assets prematurely to their plant lifetime (cf. refs. 16,17,26), so that stranded assets are computed as

$$\text{Stranded assets} = \frac{(L - R)}{L} \cdot \text{OCC} \cdot K, \qquad (1)$$

where $L$ is a power plant's standard lifetime, $R$ is a power plant's retirement age, $OCC$ is measured in US\$ per MW and $K$ represents power plant capacity in MW. We thus assume that over the power plant's lifetime productivity is constant and capital costs are recovered linearly. In computing stranded assets, we only consider plants with $L \geq R$, otherwise we define stranded assets to be zero. This approach focuses on sunk costs. It is a narrower definition of stranded assets and could thus result in a lower bound of a stranded assets estimation. We implement this approach since it requires a reduced set of assumptions and thereby facilitates an analysis of stranded assets globally. Our approach may lead to conservative estimates of stranded assets since we strand old plants with high levels of recovered capital costs first, while in reality, e.g., for geopolitical reasons, younger plants may be stranded instead—we target this in a sensitivity analysis (see the Supplementary Information). We discount stranded assets to 2021 at an interest rate of 5% (cf. ref. 16). Assumptions on power plants' standard lifetimes and OCC are provided in the Supplementary Information. In a sensitivity analysis, we alter the interest rate and power plants' standard lifetimes (please refer to the Supplementary Information).

### Data

We employ a unique combination of three data sources. First, to obtain power plants' operating capacity and a mapping from physical assets to their owners, we employ a novel data set from Asset Resolution[20]. The Asset Resolution data include multiple sectors covering more than 75% of global emissions, namely energy (fossil fuel production and power), transport (automotive, aviation, and shipping), and industry (cement and steel). 300,000 assets in these sectors are matched with 65,000 asset owners. The asset-level data contain information on technology type (e.g., energy source for power plants), asset status (e.g., active, under construction, start year), location, production, capacity, financial metrics (e.g., capital expenditures), and emission metrics (e.g., emission intensity). The ownership data allows to identify the direct owner of an asset (called "direct owner" henceforth) and its ownership tree of asset owners owning the direct owner. Along this ownership tree, each link between an asset and a direct owner is characterized by an ownership share, since an asset can be owned by multiple direct owners. Further, each link between two asset owners is

characterized by an equity share. If asset owner *A* owns a listed asset owner *B*, which issues equity securities, asset owner *A*'s equity share is the ratio of its owned shares to asset owner *B*'s total shares outstanding. If asset owner *A* owns a non-listed asset owner *B* the equity share is one. To simplify interpretation of the results, we focus on three levels of the ownership tree, namely assets, direct owners, and the asset owners at the top of each ownership tree, called "parent owner" henceforth. A parent owner may represent a (non-)listed company, a nation state or an individual shareholder. When aggregating the value of assets at the parent owner level, all ownership and equity shares along the ownership tree are accounted for.

Since this analysis focuses on power sector asset stranding, we only use the Asset Resolution data subset on the power sector. The power plant level data set covers capacity plans (decommissioned, operating, and pipeline) from 1897 to 2075 of over 135,000 unique power plants across the globe. Regarding fossil fuel power plants, the Asset Resolution data covers over 32,800 unique power plants using coal, oil, gas, or a mix of two fuels as input. The analysis requires imputing some missing information in the Asset Resolution data set (see the Supplementary Information). Please refer to the Supplementary Information for fossil fuel power plant descriptive statistics after imputing missing values.

Second, we use data from IEA's World Energy Outlook 2021[21], which provides the climate-compatible power plant capacity and the capacity following current national climate pledges. IEA uses the World Energy Model, a large-scale simulation tool, to outline development scenarios of energy demand and supply until 2050. This model covers the whole global energy system and provides projections on a sector-by-sector and region-by-region level using 2020 as a base year.

For our baseline analysis, we employ the Sustainable Development Scenario (SDS), which outlines how the global energy system can evolve in order to meet the United Nation's energy-related Sustainable Development Goals (SDG) in a cost-effective, realistic way. These goals are universal access to energy (SDG 7), reduction of severe health impacts from air pollution (part of SDG 3), and tackling climate change (SDG 13). The SDS is consistent with the 2 °C Paris goal with a 50% probability without relying on global net-negative $CO_2$ emissions. Some assumptions of the SDS are of particular relevance for our analysis. First, the SDS assumes $CO_2$ pricing differentiated between (groups of) countries. For instance, carbon prices in advanced economies with net-zero pledges start converging from 2025 on and reach US\$160/t $CO_2$ in 2050. Selected developing countries establish a $CO_2$ price that reaches US\$95/t $CO_2$ in 2050. Second, regarding power sector policies, CCUS is assumed to play a crucial role: 850 (5000) Mt of $CO_2$ emissions are captured in 2030 (2050), of which one-third is captured by the power sector, mainly in China and the US. For each region, the power capacity employing CCUS is not differentiated between fuels. To approximate how much coal and gas capacity with CCUS each region runs in each year, we multiply the share of each fuel's capacity in a region and year by its total fossil CCUS capacity.

In an additional analysis, we use the Announced Pledges Scenario (APS), which assumes that all countries implement their recently announced 2030 climate targets and longer term net-zero pledges fully and on time. Importantly, net-zero pledges can be reached by offsetting some remaining emissions from the energy sector, e.g., by absorbing emissions from forestry and land use. In comparison the the SDS, the APS highlights the ambition gap between reaching the 2 °C Paris goal and recently announced pledges. For instance, in 2050 $CO_2$ emissions from the energy sector and industrial processes reach more than 20 Gt in the APS compared to less than 10 Gt in the SDS.

Third, we retrieve data from Yahoo[27] on 338 listed parent owners' financial information. These include data on shares outstanding, market capitalization, and total equity as of 30th December 2021. Currencies are converted to US\$ using 2021 annual average exchange rates from OECD National Accounts Statistics[28] and Deutsche Bundesbank[29]. For descriptive statistics of these variables, please refer to the Supplementary Information.

## Reporting summary

Further information on research design is available in the Nature Portfolio Reporting Summary linked to this article.

## Data availability

The datasets "World Energy Outlook 2021 Extended Dataset" and "Linked Dataset (from Companies to Assets)" (vintage year Q3, 2020) are available from the International Energy Agency and Asset Resolution, respectively. Restrictions apply to the availability of these data, which were used under license for the current study and are not publicly available. Financial data on listed companies were hand-collected from the website Yahoo Finance (https://finance.yahoo.com/) on 8 July 2022 and are available from the corresponding author upon request.

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

## Acknowledgements

We gratefully acknowledge funding by the German Ministry for Education and Research under Grant No. 01LA1811C and by the Einstein Research Unit "Climate and Water under Change" from the Einstein Foundation Berlin and Berlin University Alliance under Grant No. ERU-2020-609. We thank Klaus Eisenack, Achim Hagen, Suphi Sen, and Gregor von Dulong for their support and comments on earlier drafts of the paper. Preliminary versions of the paper were presented and discussed at AURÖ young researchers Workshop 2021 in Oldenburg, FoReSee Workshop 2022 in Berlin, and EAERE Conference 2022 in Rimini. We acknowledge support by the Open Access Publication Fund of Humboldt-Universität zu Berlin.

## Author contributions

The author confirms sole responsibility for the study conception and design, data collection, analysis, interpretation of results, and manuscript preparation.

## Funding

## Competing interests

The author declares no competing interests.
