## [Peer Review File · Nature Communications]

Concentration of asset owners exposed to power sector stranded assets may trigger climate policy resistanceReviewers' Comments:

Reviewer #1:

Remarks to the Author:

Many thanks to the authors for their submission. The analysis is interesting and is well presented – although quite technical in nature it was easy to follow throughout the manuscript.

It looks to me like the analysis is highly dependent on two assumptions. Both are acknowledged in the text but I think could be expanded on further.

The first is the lifetime of the plants. I admit that 50/40/40 caused me to raise my eyebrows. My own team's modelling assumes 40 for coal and 30 for gas (oil 40). We have typically found quite low values for power sector stranded assets in a 2C scenario, but that could be because we don't go to plant level. It would be interesting to see a larger version of Table B5 with further options for plant lifetimes (even if just using the 5% interest rate).

I am not familiar with the reference given, but one possible reason for the difference is that coal plants are intended to run for 40 years but have lifetimes extended. In this case it is questionable whether the additional 10 years are part of the initial value of the plant (but worth discussing). A more minor point here is that the productivity of the plants will decrease over time (e.g. more maintenance required) especially toward the end, but I think it is OK just to acknowledge this issue.

The second point is the reliance on the IEA projections. This is more tricky. As far as I'm aware, the IEA does not go to plant level in its modelling. If correct, we cannot discount that the headline results are actually highlighting a shortcoming in the IEA's analysis, rather than real-world likely asset stranding – i.e. if the IEA went to plant level analysis would it get different results. Mercure et al (2018) discusses this a bit but requires a full techno-economic model to compare to.

In this sense the APS may be more solid, aside from any issues aggregating countries. However, this scenario may under-estimate because it assumes that countries only take climate action to meet targets. In reality, new solar power is soon going to undercut existing coal plants on a straight LCOE basis, meaning the existing plants will be rapidly phased out unless there is market intervention. Of course this gets back to the long-running discussion about the IEA's predictions of solar take-up rates.

Minor points

The abstract needs to make clear that it is referring to a 2C scenario.

The opening sentence should read 2C rather than 1.5C? There is no further mention of 1.5C.

I would perhaps tone down the references to policy. I'm not sure how clear the IEA is on policies in its scenarios but there are many different ways to reduce emissions. When the largest owners are the people that make the policy, the interpretation changes.

The climate initiatives paragraph on p18 appeared very suddenly and I still don't quite see the relevance (the assets are stranded regardless). The final sentence is more relevant, but the paragraph could be reduced.

Reviewer #2:

Remarks to the Author:

The article quantifies at the owner's level the amount of stranded assets in the power sector and trace these assets back to their final owner. The methodology of the article is to combine a dataset on ownership of power plants with data from the International Energy Agency regarding future decarbonization pathways. The analysis relies on several assumptions that are explained and documented in the article and/or in the supplementary information.

---- General appraisal----

The article is a valuable contribution to the literature. It is great to see the stranded assets literature moving in the direction of more fine grained, individual level, analyses. The article is neatly written and clearly structured (I may have missed some but I could not spot any typos. I am a bit

disappointed as I like spotting typos, but I understand not being able to spot typos is not a fair reason to ask for rejection).

I have no major contention but have nonetheless a couple of observations that may warrant a discussion/revision in the paper before it can be published.

----Remarks----

1. The authors' analysis relies on overnight capital costs. Not everyone might be familiar with this concept and the authors should therefore provide a clear definition. It is currently lacking in the article. The authors should also make clear that OCCs are not actual costs, but estimated costs if a plant were to be built overnight. OCCs are used to compare different options facing power plants owners and builders.

2. I agree with the authors that focusing on sunk costs may lead to a conservative estimate of stranded assets. But other assumptions made by the authors make it less clear whether their estimates are actually conservative:

As explained on page 6, the authors assume that "power plants are stranded entirely so that a plant's capacity cannot be partially reduced." I understand the necessity of the simplifying assumption to carry on the analysis but this is likely to inflate stranded assets estimate. It may therefore at least partly compensate for the deflation of stranded assets value due to focusing on OCCs only.

Further, if I understood correctly, the authors make the implicit assumption that OCCs are recovered over the entire standard (i.e. non stranded) lifetime of power plants. I am wondering if this is really the case though? Important infrastructures' lifetime typically exceed their amortization time. If this is also the case for power plants, the distribution of recovered OCCs would not be uniform across time but rather concentrated on the earlier years of a plant lifetime. The exact distribution of recovered OCC over time thus depends on the level of the ROI.

If recovered OCCs are concentrated on the earlier years of a plant lifetime, the value of stranded assets based on OCCs would actually be lower than the authors' estimate, because most of OCCs would be likely to be recovered by the time of stranding.

I am not totally certain I get figure 5 fully correctly (an example reading sentence in the legend could be useful to clarify the exact meaning of the figure) but using it as an example, it would mean that both companies would have a higher proportion of their operating capacity in light pink.

I acknowledge that the authors also provide a sensitivity analysis in the supplementary information. They find that a shorter standard lifetime lowers stranded assets value, which makes economic and financial sense.

To wrap up: taking the whole standard lifetime of power plants instead of the necessary lifetime for a 100% recovery of OCCs may lead to inflated stranded assets values. If data allow, the authors could try estimating OCC recovery time using power sector's ROI by weighting standard power plant lifetime by ROI and see to what extent it impacts their results.

3. There is an ambiguity in the meaning of "stranded power plant lifetime" that should be clarified as it can lead to a wrong understanding of equation 1, the formula to compute stranded assets.

If "stranded power plant lifetime" means the time after stranding, up to the date of normal decommissioning, then the equation makes sense. But then the term "lifetime" creates confusion because if the plan is stranded, it is not "alive" anymore.

However, if "stranded power plant lifetime" means the lifetime *_until_* stranding, as I understood it initially, then I think there is a problem with the equation: Indeed, as it stands now it reads stranded assets = (stranded life time / standard life time)*OCC*K. According to this meaning of "stranded assets power plant lifetime", the higher the lifetimes ratio, the higher the unrecovered OCCs. However shouldn't the ratio be the inverse, i.e., standard lifetime / stranded lifetime?

Indeed as it reads in the article, the ratio says that the closer the stranding lifetime to the standard lifetime, the higher the unrecovered OCCs. But this does not make economic sense: if the stranded lifetime of a plant is longer relatively to the standard lifetime, then the value of stranded assets should be lower because more OCCs may be recovered. And, indeed, in their sensitivity analysis, the authors do say that a shorter standard lifetime — so a higher lifetimes ratio — lowers stranded assets. Which brings me back to the beginning of remark 3: equation 1 only makes sense if "stranded power plant lifetime" means the time *_after_* stranding, up to the date of normal decommissioning. But in that case, again, I find the label "lifetime" particularly confusing.

4. As a side remark, note that your measure of stranded assets is, I think, better than measuring them through lost profits. Indeed, assets are stocks and profits are flows. Lost profits is a loss of income, not of assets. Of course, building on neoclassical capital theory, one could argue that capital value depends on the discounted value of the expected stream of income. But this is a circular reasoning because, as Sraffa and Robinson pointed out during the Cambridge controversies decades ago, the discounted value of future income depends on the interest rate, which itself, still following neoclassical capital theory, depends... on the value of capital.

It is therefore normal practice to value capital assets in balance sheets using the perpetual inventory method based on gross fixed capital formation, as done in national accounting, and which you implicitly follow using OCCs.

I am looking forward to reading the revised version.

Reviewer #3:

Remarks to the Author:

This manuscript analyzes stranded assets in the power sector at the asset level and links these to their direct and parent corporate owners. Based on the Asset Resolution database of power plant capacity, build year and owners and the IEA's power supply in its 2021 sustainable development and announced pledges scenarios by fuel and region, the regional stranding is computed as the the share of years of the operational lifetime of a plant that it doesn't run in order not to exceed IEA supplies time capital expenditure. The stranded assets' owners, geographical distribution and the correlation of their stranded assets with investments in low-carbon technologies are analyzed. A number of results are highlighted that show, among others, the regionally different size of total stranded assets and the nature of stranded asset owners.

This is a careful empirical work with an intuitive design that uses the widely implemented IEA scenarios as a widely accepted guideline for regional future supply and a novel power sector asset-level dataset. The analysis of the incidence of stranded assets and who are their owners can be important for designing policies to avoid social repercussions or, as the authors stress, anticipate resistance to climate policies. Perhaps more abstractly, it is also helpful to understand that the overall stranding in the global power sector over decades may amount to no more than a few hundreds of billions of dollars, not even 1% of GDP in one year and could thus be easily compensated if countries could coordinate. I enjoyed learning about these issues and seeing the quantitative results very much.

While thus a very useful contribution to knowledge in general, I found the paper conspicuously lacking a clear focus. In order to be publishable, it would need to actually ask a clear, focused question and

provide an answer. In this version, I could not quite detect this (or perhaps the authors simply weren't clear in communicating it). Even in the abstract there is no 'here we show' but instead 'a dataset is presented' which is 'employed to analyze'. But what is the question, a 'thorough understanding' is not specific enough nor well motivated. In the paper, this lack of focus carries through and the reader gets a series of graphs whose significance is not immediately clear. The authors need to be much clearer what the main message is. That is not because there isn't one. In fact, at present, there are a lot of stories in there that aren't developed. Can you pick one or two? What results to focus on? And if the authors pick, e.g., the regional distribution as the focal point, I would like to see some sensitivity analysis, like when maximal stranding is assumed in rich countries? Or minimum? What about the possibility/or absence of CCS? If they instead focus on the parent owners type, then what is the significance, why should I care whether it's public or private (better listed or non-listed to avoid confusion with state owned) etc.? Does it have different consequences for whether there is resistance to climate policies? If not, why are these results shown? Without understanding the focus, it is hard for me to comment on specifics, so I ask the authors to provide that focus.

Some other, smaller comments:

What about the discount rate used (if any)? It's not a lot of money, especially over time and what effect has the discount rate here? Relatedly, what if the accounting time is shorter than operational lifetime? 50 years sounds awfully long for recuperating an investment, but perhaps that's how long it takes for investors to get their return. Can the authors say more about how long a coal power plant must run to be depreciated?

As far as I know for other Nature journals, conclusions don't repeat what's been said before. Overall, the structure could be checked for content guidelines for the journal.

"For simplicity we assume that power plants are always stranded entirely, so that a plant's capacity cannot be partially reduced." ... How strong of an assumption is this?

Figure 8: Why are just the top 20 owners displayed? Why not all? Can you make it as some sort of % rather than total values or log scales so that small sizes remain visible? In general for plots I found those with all the data, rather than just the top x more insightful. (There are many plots; is there an upper limit at the journal?)

A couple of additional reflections on significance of content/focus, which perhaps help with thinking about my main comment above:

Better distinguish state from non-state owners (also in graphs). Seems like the public/private distinction doesn't really give much analytical mileage from just considering the graphs.

Figure 7: "Listed parent owners with negative total equity or stranded assets larger than their total equity are ignored." ... Why? Also, they are some of the most interesting, being technically insolvent, no?

Does the Republic of China really care whether it owns a few stranded assets according to IEA? Doesn't it rather have strategic spare capacity? Isn't private owners much more interesting?

So the big losers are Indonesia (perhaps Vietnam) and India. What does this mean?

Response to Reviewer #1

Dear colleague,

Thank you for your engagement with our manuscript. We include below the entirety of your response and address your concerns point-by-point.

Many thanks to the authors for their submission. The analysis is interesting and is well presented – although quite technical in nature it was easy to follow throughout the manuscript.

It looks to me like the analysis is highly dependent on two assumptions. Both are acknowledged in the text but I think could be expanded on further.

Thank you for your kind evaluation. We seek to address your suggestions in what follows.

1. The first is the lifetime of the plants. I admit that 50/40/40 caused me to raise my eyebrows. My own team's modelling assumes 40 for coal and 30 for gas (oil 40). We have typically found quite low values for power sector stranded assets in a 2C scenario, but that could be because we don't go to plant level. It would be interesting to see a larger version of Table B5 with further options for plant lifetimes (even if just using the 5% interest rate). I am not familiar with the reference given, but one possible reason for the difference is that coal plants are intended to run for 40 years but have lifetimes extended. In this case it is questionable whether the additional 10 years are part of the initial value of the plant (but worth discussing). A more minor point here is that the productivity of the plants will decrease over time (e.g. more maintenance required) especially toward the end, but I think it is OK just to acknowledge this issue.

Thank you for this comment. We agree that the assumptions on plant lifetimes should be elaborated given the various lifetimes for fossil fuel power plants in the literature. We have amended the paper as follows: We now explain in the supplementary information on p. 1 (blue font), that we would overestimate stranded assets if we used extended lifetimes rather than those that plants are intended to run for initially:

“Power plants’ OCC may recover over shorter periods, in which case we could overestimate stranded assets. Thus, we alter power plant standard lifetimes in a sensitivity analysis.”

Further, we extended Table 5 to include a range of shorter plant lifetimes at a discount rate of 3%, 5%, and 7%, respectively (see supplementary information, p. 6). The table also differentiates between fossil fuel plant types to increase transparency of the estimates. Results show that for lifetimes of 40 years for coal, 30 for gas, and 40 for oil, respectively, and an interest rate of 5% we have stranded assets as high as US\$277, 21, and 15 billion, for coal, gas, and oil, respectively. Regarding the decrease in a plant’s productivity over the lifetime, we agree that this issue should be acknowledged in the manuscript. We have added a remark in the Methods section (see footnote 11, p. 19, blue font):

“We thus assume that over the power plant’s lifetime productivity is constant and capital costs are recovered linearly.”

2. The second point is the reliance on the IEA projections. This is more tricky. As far as I’m aware, the IEA does not go to plant level in its modelling. If correct, we cannot discount that the headline results are actually highlighting a shortcoming in the IEA’s analysis, rather than real-world likely asset stranding – i.e. if the IEA went to plant level analysis would it get different results. Mercure et al (2018) discusses this a bit but requires a full techno-economic model to compare to. In this sense the APS may be more solid, aside from any issues aggregating countries. However, this scenario may under-estimate because it assumes that countries only take climate action to meet targets. In reality, new solar power is soon going to undercut existing coal plants on a straight LCOE basis, meaning the existing plants will be rapidly phased out unless there is market intervention. Of course this gets back to the long-running discussion about the IEA’s predictions of solar take-up rates.

Thank you for this comment and for pointing out the paper by Mercure et al. (2018), which is highly relevant to our study. We agree that a limitation of our study is its reliance on IEA projections as their assumptions eventually affect our valuation of stranded assets. To be able to perform an estimation of stranded assets, however, we have to use a projection and we think that the IEA projections are most widely implemented despite their shortcomings. In the discussion,

we acknowledge and discuss this limitation. We added to the discussion on p. 17 (blue font) that the diffusion of low-carbon technologies may affect asset stranding as shown in Mercure et al. (2018):

“There are some limitations to this analysis. First, the IEA scenario assumptions on climate policies, prices, technological progress, behavior, etc. are crucial for the valuation of stranded assets. For instance, the stringency and the design of policies can affect the distribution and absolute value of stranded assets. Energy efficiency improvements at the plant level, advances in CCUS retrofitability and net-negative emission technologies could reduce stranded assets. On the flip-side, changes in preferences and the diffusion of low-carbon technologies may increase stranded assets (Mercure et al., 2018).”

3. Minor points: The abstract needs to make clear that it is referring to a 2C scenario. The opening sentence should read 2C rather than 1.5C? There is no further mention of 1.5C.

Thank you for the suggestion to clarify that we are referring to a 2° C scenario. The first sentence in the abstract now incorporates this and reads (p. 1):

“Thoroughly assessing the owners and distribution of stranded assets in a 2° C scenario is essential to anticipate climate policy resistance.”

Thank you for pointing out this inconsistency. We have changed the first sentence of the introduction (p. 2) to read:

“Reaching the 2° C climate goal requires the implementation of stringent policies to transform the energy sector.”

4. I would perhaps tone down the references to policy. I’m not sure how clear the IEA is on policies in its scenarios but there are many different ways to reduce emissions. When the largest owners are the people that make the policy, the interpretation changes.

Thank you for this comment. As outlined in the Methods section on p. 22 the IEA assumes CO₂ price paths differentiated by developed and developing countries. The focus on policy is central to motivate our research: Identification of the asset owners is key to anticipating policy resistance and providing feasible policy advice. This is also crucial if asset owners are national governments, who make policies, as they will represent their interests in international (climate) negotiations. However, we agree that emission reduction can also be the result of other means such as low-carbon technology diffusion or changing social preferences. We have amended the paper to acknowledge this point in the introduction (see footnote 1 on p. 2, blue font):

“Assets may also strand due to climate impacts or transition risks, which are not directly linked to climate policies such as changing social preferences.”

5. The climate initiatives paragraph on p18 appeared very suddenly and I still don't quite see the relevance (the assets are stranded regardless). The final sentence is more relevant, but the paragraph could be reduced.

Thank you for this remark. We agree that this paragraph does not smoothly add to the storyline of the paper. However, investor engagement related to asset stranding has recently gained attention in research (cf. von Schickfus, 2021, ifo Working Paper No. 356), which is why we have performed this additional analysis. To keep these additional results in the paper without compromising on the flow of reading, we have moved this paragraph to the supplementary information (see “Shareholder engagement” on p. 6 of the supplementary information).

Response to Reviewer #2

Dear colleague,

Thank you for your engagement with our manuscript. We include below the entirety of your response and address your concerns point-by-point.

The article quantifies at the owner's level the amount of stranded assets in the power sector and trace these assets back to their final owner. The methodology of the article is to combine a dataset on ownership of power plants with data from the International Energy Agency regarding future decarbonization pathways. The analysis relies on several assumptions that are explained and documented in the article and/or in the supplementary information.

---- General appraisal----

The article is a valuable contribution to the literature. It is great to see the stranded assets literature moving in the direction of more fine grained, individual level, analyses. The article is neatly written and clearly structured (I may have missed some but I could not spot any typos. I am a bit disappointed as I like spotting typos, but I understand not being able to spot typos is not a fair reason to ask for rejection).

I have no major contention but have nonetheless a couple of observations that may warrant a discussion/revision in the paper before it can be published.

Thank you for your kind evaluation. We seek to address your suggestions in what follows.

1. The authors' analysis relies on overnight capital costs. Not everyone might be familiar with this concept and the authors should therefore provide a clear definition. It is currently lacking in the article. The authors should also make clear that OCCs are not actual costs, but estimated costs if a plant were to be built overnight. OCCs are used to compare different options facing power plants owners and builders.

Thank you for this comment. We agree that the reader may not be familiar with the concept and have added a definition, where we first mention OCC (see footnote 2 in the introduction, p. 4, blue font):

“OCC cover a power plant's pre-construction, construction, and contingency costs but exclude interest during its construction -- as if the plant was built overnight (International Energy Agency and Nuclear Energy Agency, 2020). OCC are used to evaluate and assess different power plant project options (Kooimey and Hultman, 2007).”

2. I agree with the authors that focusing on sunk costs may lead to a conservative estimate of stranded assets. But other assumptions made by the authors make it less clear whether their estimates are actually conservative: As explained on page 6, the authors assume that “power plants are stranded entirely so that a plant’s capacity cannot be partially reduced.” I understand the necessity of the simplifying assumption to carry on the analysis but this is likely to inflate stranded assets estimate. It may therefore at least partly compensate for the deflation of stranded assets value due to focusing on OCCs only. Further, if I understood correctly, the authors make the implicit assumption that OCCs are recovered over the entire standard (i.e. non stranded) lifetime of power plants. I am wondering if this is really the case though? Important infrastructures’ lifetime typically exceed their amortization time. If this is also the case for power plants, the distribution of recovered OCCs would not be uniform across time but rather concentrated on the earlier years of a plant lifetime. The exact distribution of recovered OCC over time thus depends on the level of the ROI. If recovered OCCs are concentrated on the earlier years of a plant lifetime, the value of stranded assets based on OCCs would actually be lower than the authors’ estimate, because most of OCCs would be likely to be recovered by the time of stranding. I am not totally certain I get figure 5 fully correctly (an example reading sentence in the legend could be useful to clarify the exact meaning of the figure) but using it as an example, it would mean that both companies would have a higher proportion of their operating capacity in light pink. I acknowledge that the authors also provide a sensitivity analysis in the

supplementary information. They find that a shorter standard lifetime lowers stranded assets value, which makes economic and financial sense. To wrap up: taking the whole standard lifetime of power plants instead of the necessary lifetime for a 100% recovery of OCCs may lead to inflated stranded assets values. If data allow, the authors could try estimating OCC recovery time using power sector's ROI by weighting standard power plant lifetime by ROI and see to what extent it impacts their results.

Thank you for this comment. As you correctly point out, the assumption on partial decommissioning and on the recovery time and distribution of OCC are relevant to our estimate of stranded assets. Allowing for partial decommissioning and reducing the recovery time may deflate our estimate. We have amended the paper by toning down on our estimate to be conservative. In the methods part the respective part in the manuscript now reads (pp. 19-20, see blue font for changes):

“This approach focuses on sunk costs. It is a narrower definition of stranded assets and could thus result in a lower bound of a stranded assets estimation. We implement this approach since it requires a reduced set of assumptions and thereby facilitates an analysis of stranded assets globally. Our approach may lead to conservative estimates of stranded assets since we strand old plants with high levels of recovered capital costs first, while in reality, e.g. for geopolitical reasons, younger plants might be stranded instead – we target this in a sensitivity analysis.”

In the discussion, the respective part now reads (p. 17, see blue font for changes):

“Second, this study assesses stranded assets as sunk costs. Stranded assets in terms of lost profits may be a lot higher with important implications for the feasibility of climate policies.”

Regarding Figure 6 (Figure 5 in the previous manuscript), we agree that the figure is not straightforward to understand. Thank you for the suggestion to include an example reading in the legend. We have now included one such sentence, which reads (p. 12):

“Example reading of the legend: This color [light pink] represents stranded capacity of which more than 90% of OCC were already recovered at the time of stranding.”

Thank you for your suggestion to estimate OCC recovery time using the power sector's ROI. We think this is a great idea to address the issue that OCC may not be recovered linearly over the entire plant lifetime but instead be concentrated in the earlier years. However, we could not find trustworthy data for the power sector's ROI. This may explain why the approach you suggest has not been implemented in the literature. To address this issue regardless, we have extended Table 5 to include a wider range of shorter plant lifetimes at various discount rates (see supplementary information, p. 6). The table also differentiates between fossil fuel plant types to increase transparency of the estimates. We hope that these more fine-grained results help to approach the scenario with OCC recovered over a shorter time or concentrated in the early years after plant commissioning.

3. There is an ambiguity in the meaning of “stranded power plant lifetime” that should be clarified as it can lead to a wrong understanding of equation 1, the formula to compute stranded assets. If “stranded power plant lifetime” means the time after stranding, up to the date of normal decommissioning, then the equation makes sense. But then the term “lifetime” creates confusion because if the plan is stranded, it is not “alive” anymore. However, if “stranded power plant lifetime” means the lifetime until stranding, as I understood it initially, then I think there is a problem with the equation: Indeed, as it stands now it reads $\text{stranded assets} = (\text{stranded life time} / \text{standard life time}) * \text{OCC} * K$. According to this meaning of “stranded assets power plant lifetime”, the higher the lifetimes ratio, the higher the unrecovered OCCs. However shouldn't the ratio be the inverse, i.e., $\text{standard lifetime} / \text{stranded lifetime}$? Indeed as it reads in the article, the ratio says that the closer the stranding lifetime to the standard lifetime, the higher the unrecovered OCCs. But this does not make economic sense: if the stranded lifetime of a plant is longer relatively to the standard lifetime, then the value of stranded assets should be lower because more OCCs may be recovered. And, indeed, in their sensitivity analysis, the authors do say that a shorter standard lifetime — so a higher lifetimes ratio — lowers stranded assets. Which brings me back to the beginning of remark 3: equation 1 only makes sense if “stranded power plant lifetime” means the time after stranding, up to the date of normal

decommissioning. But in that case, again, I find the label “lifetime” particularly confusing.

Thank you for pointing out this ambiguity. We completely agree and have amended the paper to clarify the meaning of Equation 1 (see Methods section, p. 19, see blue font for changes):

“stranded assets are computed as

$$\text{Stranded assets} = ((L-R)/L) * OCC * K,$$

where L is a power plant's standard lifetime, R is a power plant's retirement age, OCC is measured in US\$ per MW and K represents power plant capacity in MW. In computing stranded assets, we only consider plants with $L \geq R$, otherwise we define stranded assets to be zero.”

We have also changed the label “lifetime” to “standard lifetime” elsewhere in the manuscript (see blue font for changes: Methods, p. 20; Supplementary information, p. 1 and Table 5 on p. 6).

4. As a side remark, note that your measure of stranded assets is, I think, better than measuring them through lost profits. Indeed, assets are stocks and profits are flows. Lost profits is a loss of income, not of assets. Of course, building on neoclassical capital theory, one could argue that capital value depends on the discounted value of the expected stream of income. But this is a circular reasoning because, as Sraffa and Robinson pointed out during the Cambridge controversies decades ago, the discounted value of future income depends on the interest rate, which itself, still following neoclassical capital theory, depends... on the value of capital. It is therefore normal practice to value capital assets in balance sheets using the perpetual inventory method based on gross fixed capital formation, as done in national accounting, and which you implicitly follow using OCCs.

Thank you for confirming the appropriateness of our stranded assets measure.

Response to Reviewer #3

Dear colleague,

Thank you for your engagement with our manuscript. We include below the entirety of your response and address your concerns point-by-point.

This manuscript analyzes stranded assets in the power sector at the asset level and links these to their direct and parent corporate owners. Based on the Asset Resolution database of power plant capacity, build year and owners and the IEA's power supply in its 2021 sustainable development and announced pledges scenarios by fuel and region, the regional stranding is computed as the the share of years of the operational lifetime of a plant that it doesn't run in order not to exceed IEA supplies time capital expenditure. The stranded assets' owners, geographical distribution and the correlation of their stranded assets with investments in low-carbon technologies are analyzed. A number of results are highlighted that show, among others, the regionally different size of total stranded assets and the nature of stranded asset owners.

This is a careful empirical work with an intuitive design that uses the widely implemented IEA scenarios as a widely accepted guideline for regional future supply and a novel power sector asset-level dataset. The analysis of the incidence of stranded assets and who are their owners can be important for designing policies to avoid social repercussions or, as the authors stress, anticipate resistance to climate policies. Perhaps more abstractly, it is also helpful to understand that the overall stranding in the global power sector over decades may amount to no more than a few hundreds of billions of dollars, not even 1% of GDP in one year and could thus be easily compensated if countries could coordinate. I enjoyed learning about these issues and seeing the quantitative results very much.

Thank you for your kind evaluation. We seek to address your suggestions in what follows.

1. While thus a very useful contribution to knowledge in general, I found the paper conspicuously lacking a clear focus. In order to be publishable, it would need to actually ask a clear, focused question and provide an answer. In this version, I could

not quite detect this (or perhaps the authors simply weren't clear in communicating it). Even in the abstract there is no 'here we show' but instead 'a dataset is presented' which is 'employed to analyze'. But what is the question, a 'thorough understanding' is not specific enough nor well motivated. In the paper, this lack of focus carries through and the reader gets a series of graphs whose significance is not immediately clear. The authors need to be much clearer what the main message is. That is not because there isn't one. In fact, at present, there are a lot of stories in there that aren't developed. Can you pick one or two? What results to focus on? And if the authors pick, e.g., the regional distribution as the focal point, I would like to see some sensitivity analysis, like when maximal stranding is assumed in rich countries? Or minimum? What about the possibility/or absence of CCS? If they instead focus on the parent owners type, then what is the significance, why should I care whether it's public or private (better listed or non-listed to avoid confusion with state owned) etc.? Does it have different consequences for whether there is resistance to climate policies? If not, why are these results shown? Without understanding the focus, it is hard for me to comment on specifics, so I ask the authors to provide that focus.

Thank you for this comment. We agree that we were not clear enough in communicating our research questions and focusing on them. To anticipate resistance to climate policies, the paper aims at identifying the owners of stranded assets as well as the extent and patterns of asset stranding. As asset stranding can be less concerning if owners are also invested in alternative energy assets, the study further analyzes ownership of stranded assets in conjunction with alternative energy assets. Against this background, we think it is important to identify both, regions/countries and owners below the country level affected by asset stranding: While regions'/countries' exposure to stranded assets may be relevant to anticipate policy resistance at international negotiations, (non-)listed owners are equally important as they can oppose policies at the national level. For listed owners, we additionally relate stranded assets to their share prices and equity as an alternative measure for their exposure to asset stranding, relevant for their potential resistance to policies.

We have implemented several changes to improve the communication and the focus of our research questions: First, we have revised the abstract to motivate and state our central research question and introduce our results (see blue font in the abstract, p. 1):

“Thoroughly assessing the owners and distribution of stranded assets in a 2° C scenario is essential to anticipate climate policy resistance. We employ novel data to analyze owners and incidence of asset stranding in the power sector globally. We show that [...]”

Second, in the introduction we have added a paragraph to motivate and state our research questions (see blue font in the introduction, p. 2):

“To assess potential sources of resistance to climate policies and to gain a better understanding of who has high stakes in national policy formation and international climate negotiations we ask: Who are the owners of power sector stranded assets across the globe and how are stranded assets distributed between them? Further, resistance to climate policies may be moderated if affected asset owners are also invested in alternative energy assets - potentially even benefiting from these policies. Thus, we ask whether asset owners' ownership of alternative energy assets correlates with asset stranding exposure.”

Third, throughout the results part of the paper we have added several sentences motivating how the results shown relate to our research questions (see blue font). Further, we have removed the last results chapter “The stranded assets ambition gap” and instead integrated the findings into the first chapter of the results, now called “Stranded assets across regions and fuels”. We agree that the chapter “The stranded assets ambition gap” did not clearly fit the storyline. The new chapter “Stranded assets across regions and fuels” shows regions, which may face strong resistance to policies as they are yet to raise the stringency of their climate targets. Finally, we agree that the paragraph on investor engagement and climate initiatives (see p. 18 in the previous manuscript) does not smoothly add to the storyline of the paper. However, investor engagement related to asset stranding has recently gained attention in research (cf. von Schickfus, 2021, ifo Working Paper No. 356), which is why we have performed this additional analysis. To keep these additional results in the paper without compromising on the flow of reading, we have moved this paragraph to the supplementary information (see section “Shareholder engagement” on p. 6). In both, the abstract

and the introduction, we have removed some of the results and specified other results to be in line with the changes in the results part of the paper.

Regarding sensitivity analyses, thank you for the great suggestion to analyze asset stranding according to a “stranding rule” different from stranding old plants first. We have now performed additional analyses assuming either maximal or minimal stranding in rich countries (see section “Sensitivity analyses” on pp. 5-6 in the supplementary information, blue font). Results show that maximal (minimal) stranding in rich countries increases stranded assets to US\$632 (657) billion:

“[...] Further, the stranded assets quantification depends on which power plant is stranded first, whenever the operating capacity exceeds the climate-compatible capacity. In our baseline analysis we strand the oldest power plants in a given region first. This approach aims at minimizing unrecovered capital costs and thus stranded assets. One could argue, however, that it is more realistic to strand power plants in the richest or poorest country in a given region first. We implement this alternative approach and rank power plants to be stranded in a given region by GDP per capita [current US\$] in 2020 in the country the plant is located. The plant's age remains only relevant in case several plants are located in the same country. Results show that stranded assets increase to US\$657 (632) billion if plants located in poor (rich) countries in a given region are stranded first.”

Thank you for the idea to perform a sensitivity analysis with respect to the possibility or absence of CCUS technologies. As mentioned in the Methods section on p. 22, CCUS is assumed in the IEA scenarios:

“850 (5000) Mt of CO₂ emissions are captured in 2030 (2050), of which one-third is captured by the power sector, mainly in China and the US.”

Performing a sensitivity analysis with respect to the possibility of CCS is unfortunately hardly feasible because power plant capacity using CCS is used as an input to the IEA projections. If we removed this capacity from our analysis, we would not be able to guarantee that enough power capacity is globally available. This could for instance violate the Sustainable Development Goal 7 (universal access to energy), that the IEA’ Sustainable Development Scenario aims at fulfilling.

Thank you for pointing out that public companies can be confused with state-owned companies. In Figure 3 on p. 9, we now call public and private companies listed and non-listed companies, respectively. We have kept the differentiation between listed and non-listed companies in this figure as it demonstrates that many listed companies are among the top most exposed entities and thus it motivates diving into the asset stranding exposure of this asset owner type in more detail. To convey this motivation to the reader we have added an introductory sentence to the results chapter “Stranded assets owned by listed parent owners” (see p. 13 in the results, blue font):

“As depicted in Figure 3, listed asset owners in various countries are among the top most exposed entities. These owners may oppose climate policies if resulting stranded assets depress their share prices.”

Some other, smaller comments:

2. What about the discount rate used (if any)? It's not a lot of money, especially over time and what effect has the discount rate here? Relatedly, what if the accounting time is shorter than operational lifetime? 50 years sounds awfully long for recuperating an investment, but perhaps that's how long it takes for investors to get their return. Can the authors say more about how long a coal power plant must run to be depreciated?

Thank you for this remark. We use a discount rate of 5% (see the Methods section on p. 20). In a sensitivity analysis we vary this discount rate to be 3 and 7% respectively (see “Sensitivity analyses” section on p. 5 and Table 5 on p. 6 in the supplementary information). Results show that, for instance, increasing the discount rate by 2 percentage points (from 3 to 5%) decreases coal stranded assets by US\$93 billion.

We agree that the assumptions on plant lifetimes should be elaborated given the various lifetimes for fossil fuel power plants in the literature. We have amended the paper as follows: We now explain in the supplementary information on p. 1 (blue font), that we would overestimate stranded assets if capital costs could be recovered over a period shorter than the operational lifetime:

“Power plants’ OCC may recover over shorter periods, in which case we could overestimate stranded assets. Thus, we alter power plant standard lifetimes in a sensitivity analysis.”

Further, we extended Table 5 to include a range of shorter plant lifetimes at a discount rate of 3%, 5%, and 7%, respectively (see supplementary information, p. 6). The table also differentiates between fossil fuel plant types to increase transparency of the estimates.

3. As far as I know for other Nature journals, conclusions don't repeat what's been said before. Overall, the structure could be checked for content guidelines for the journal.

Thank you for this comment. We have removed the summary of the results from the discussion and adjusted the structure of the paper according to the journal guidelines. In particular, this includes restructuring the introduction and shortening both the title and the abstract.

4. "For simplicity we assume that power plants are always stranded entirely, so that a plant's capacity cannot be partially reduced." ... How strong of an assumption is this?

Thank you for this remark. If a plant's capacity would be partially reduced in one year, its remaining capacity would be the first to be stranded in the consecutive year(s) with climate compatible capacities decreasing over time. Stranding the remaining capacities in the future instead of today, would decrease their present value as we discount stranded assets to 2021. Thus, the value of stranded assets resulting from relaxing this assumption could be lower compared to the approach implemented. To acknowledge this potential inflation of our stranded assets estimates, we have toned down on our estimate to be conservative. In the Methods part the respective part in the manuscript now reads (pp. 19-20, see blue font for changes):

"This approach focuses on sunk costs. It is a narrower definition of stranded assets and could thus result in a lower bound of a stranded assets estimation. We implement this approach since it requires a reduced set of assumptions and thereby facilitates an analysis of stranded assets globally. Our approach may lead to conservative estimates of stranded assets since we strand old plants with high levels of recovered capital costs first, while in reality, e.g. for geopolitical reasons, younger plants might be stranded instead – we target this in a sensitivity analysis."

In the discussion, the respective part now reads (p. 17, see blue font for changes):

"Second, this study assesses stranded assets as sunk costs. Stranded assets in terms of lost profits may be a lot higher with important implications for the feasibility of climate policies."

5. Figure 8: Why are just the top 20 owners displayed? Why not all? Can you make it as some sort of % rather than total values or log scales so that small sizes remain visible? In general for plots I found those with all the data, rather than just the top x more insightful. (There are many plots; is there an upper limit at the journal?)

Thank you for this remark and the valuable suggestion. We agree and have amended the paper as follows:

In Figure 8 on p. 16 of the results part, Panel (b), we have changed the scales on both axes to log-scales and included all observations instead of only focusing on the top 20 owners. We think the plot is now more informative conveying the positive correlation between asset stranding exposure and alternative energy ownership mentioned in the manuscript.

In Figure 7 (Figure 6 in the previous manuscript) on p. 14 of the results part, Panel (a) and (b), we have changed the plot type from showing the top 30 most exposed listed parent owners measured by stranded assets per share outstanding [US\$] and stranded assets in share price [%] to boxplots, so that they now show all listed asset owners. We think this plot type is more informative, as it shows additional information, such as the quartiles and medians of the two measures differentiated by OECD and non-OECD countries, which support the points we make in the manuscript (stronger exposure by OECD listed asset owners compared to non-OECD owners, but higher ratio of stranded assets to equity for non-OECD owners).

We think that showing the distribution of stranded assets among the top losers in Figures 3 and 4 (previously Figure 2 and 3) on p. 9 and 10 of the results part is highly illustrative and relevant for the storyline: As explained in the manuscript, the formation of resistance to policies (most relevant for the most exposed owners) may be easier if stranded assets are concentrated in a small number of asset owners. We have tried different plot types, e.g. boxplot, which would allow showing all asset owners. However, this made comparison of the distribution of stranded assets of the top losers across countries (Figure 3) and across direct/parent owner (Figure 4) more difficult and did not reveal additional insights.

We doubled-checked for the upper limit of number of plots (10), which we do not exceed.

A couple of additional reflections on significance of content/focus, which perhaps help with thinking about my main comment above:

6. Better distinguish state from non-state owners (also in graphs). Seems like the public/private distinction doesn't really give much analytical mileage from just considering the graphs.

Thank you for this remark. We agree that the distinction between state and non-state owners would be highly interesting in the context of stranded assets and climate policies. Unfortunately, however, we do not have this information in the data consistently, especially in mixed-form ownerships between state and non-state owners. At the same time, we think that the listed/non-listed owner distinction in Figure 3 on p. 9 of the results part is highly valuable: It demonstrates that listed asset owners are among the top most exposed entities and thus it motivates analyzing stranded assets owned by listed owners in more detail (see our response to your main point above).

7. Figure 7: "Listed parent owners with negative total equity or stranded assets larger than their total equity are ignored." ... Why? Also, they are some of the most interesting, being technically insolvent, no?

Thank you for this comment. We agree that these owners are highly interesting and worth displaying and we have added them to the plots (see Figure 7, Panel (c), on p. 14 of the Results part and Figure 10 on p. 7 of the supplementary information). We decided, however, not to discuss these owners in more detail for two reasons: First, these owners present only a small group in our sample ($n=8$) so that we do not want to draw broad conclusions from these observations. Second, these owners are only technically insolvent. Their cash flow, which we unfortunately have no data on, could be enough to cover their debt and enable them to keep up operations.

8. Does the Republic of China really care whether it owns a few stranded assets according to IEA? Doesn't it rather have strategic spare capacity? Isn't private owners much more interesting?

Thank you for this remark. We agree that looking at asset stranding at the regional or country level only provides limited insights into potential social repercussions and climate policy resistance. This is why this study goes beyond the regional/country level and also analyzes stranded assets owned by (non-)listed companies. It would also be interesting to distinguish between public and private owners, however, as explained above (see our response to your point 6), the data unfortunately does not allow this consistently.

9. So the big losers are Indonesia (perhaps Vietnam) and India. What does this mean?

Thank you for suggesting elaborating on this finding. We agree it is worth discussing that India and Asia-Pacific excluding China, India, and Japan are those regions strongly exposed to stranded assets while owning little to moderate degrees of alternative energy capacity. We think that these regions may strongly resist implementing climate policies domestically and making climate pledges at international negotiations. We have added this thought to the results section “Stranded assets and alternative energy assets” (see blue font on p. 16):

“[...] In contrast, India and Asia-Pacific excluding China, India, and Japan may strongly resist the implementation of climate policies or the announcement of stringent climate pledges – an avenue for future research with high policy relevance.”

Reviewers' Comments:

Reviewer #1:

Remarks to the Author:

Many thanks to the authors for their responses and revisions. I would like to come back briefly on a few points.

Normative comments about countries' sustainability. I don't think many people would agree that the US has fully sustainable policies but Europe does not - the reality is that the US has a long way to catch up. This appears to be a misinterpretation of the IEA's scenarios, which I assume push Europe further in the SDS *because* it has already done much of the groundwork. I would therefore rephrase the text here and revise the interpretation.

IEA limitations on p17. I would stress that the key limitation in the IEA analysis is that it does not go to plant level (as far as I know) and they might have different results if they did. I am still not completely convinced that a large reported volume of stranded assets is not just a reflection of shortcomings in the IEA projections. Maybe a footnote to make that point would cover it.

Governments likely to negotiate harder if they own the assets. This conclusion treats government as if it was a private company, which I'm not sure is correct. Governments should be representing all their citizens, including business owners, in negotiations. In many ways it is easier if stranded assets are public - the risk of default is much lower. The people representing the government would not normally personally profit from publicly owned assets?

Reviewer #2:

Remarks to the Author:

I would like to thank the authors for their effort to engage with the reviewers' comments and revise their manuscript.

As far as my comments are concerned, I am satisfied with the treatment of the authors. They provided convincing revisions and replies. I also acknowledge their explanations when some of my suggestions could not be implemented due to lack of data (e.g. weighting lifetime with ROI).

I therefore recommend publication.

Reviewer #3:

Remarks to the Author:

I congratulate the authors to this very effective revision of their initial submission. To me, the stronger focus on resistance by asset owners is compelling, and the paper now a really engaging read. I have a few minor comments.

- 1) Consider changing your title to reflect the 'resistance' aspect. 'Disentangling' does not convey a lot of information. You could rather highlight the analysis of potential resistance; and even write something along the lines of "Concentration of asset owners may lead to resistance or some such". I imagine that this would increase the potential impact your interesting paper has.
- 2) In the abstract, you write owners "may face stranded assets of almost 78%", but looking at your (excellently revised) box plots in figure 7, isn't that the upper limit? Consider changing slightly to "assets of *up to* 78%".
- 3) In footnote 1, you could also cite Semieniuk et al. (2021) as a thorough discussion of transition risks.

4) In the introduction you first only refer to 2degC and then to 1.5 and 2 both on page 1. Can you be consistent one way or the other?

5) Page 4 bottom, "in contrast to the US" – it is a bit unclear what is being contrasted here. Does the US have a pledge? If it does, what does it not fall short of. I feel an additional sentence setting the stage for this one would help communicate your point.

6) P. 7, line 105, "than those currently announced" sounds like the stranded assets are announced. Surely you're referring to policies? Do you mean: "than those under currently announced policies?"

7) P. 15 l208, I suggest striking 'ignored' and instead simply writing "or 80% for those owners where stranded assets do not exceed equity". (I am not sure how negative equity is relevant to this, as it would result in a negative number).

8) The revised Figure 8 is great. Personally I like showing the actual numbers behind the logs better (the softwares I know allow plotting the original numbers but specifying a 'log scale'), since it's not necessarily clear what log 18 really translates to. But at a minimum, the base of the log should be given in the caption.

Sincerely,

Gregor Semieniuk (journal encourages signing reports)

Literature mentioned

Semieniuk, G., Campiglio, E., Mercure, J.-F., Volz, U., Edwards, N.R., 2021. Low-carbon transition risks for finance. *WIREs Clim. Chang.* 12, e678.

Response to Reviewer #1

Dear colleague,

Thank you for your engagement with our manuscript. We include below the entirety of your response and address your concerns point-by-point.

Many thanks to the authors for their responses and revisions. I would like to come back briefly on a few points.

Thank you for your kind evaluation. We seek to address your suggestions in what follows.

1. Normative comments about countries' sustainability. I don't think many people would agree that the US has fully sustainable policies but Europe does not - the reality is that the US has a long way to catch up. This appears to be a misinterpretation of the IEA's scenarios, which I assume push Europe further in the SDS *because* it has already done much of the groundwork. I would therefore rephrase the text here and revise the interpretation.

Thank you for this comment. We agree that most people would not agree on the US having implemented fully sustainable policies. The IEA's Announced Pledges Scenario, however, does not incorporate *implemented* policies but those that are *announced*. Since the US pledged achieving a net-zero emissions economy by 2050, fossil power capacities in its Sustainable Development Scenario and Announced Pledges Scenario are in line. This also holds for the EU, which also pledged net-zero by 2050. In our analysis, however, we consider *Europe* and not the *EU*. Thus, for Europe the two scenarios are not aligned, which is driven by European countries outside the EU. To avoid confusing the reader and conveying the impression of normative comments we have removed comments on the US in this context and clarified that announced pledges from European countries outside the EU fall short of those required for a sustainable development (see lines 66-70; 120-123).

2. IEA limitations on p17. I would stress that the key limitation in the IEA analysis is that it does not go to plant level (as far as I know) and they might have different results if they did. I am still not completely convinced that a large reported volume of stranded assets is not just a reflection of shortcomings in the IEA projections. Maybe a footnote to make that point would cover it.

Thank you for this comment. The IEA indeed does not go to the plant level. We agree that this could alter their results and in turn change out estimates of stranded assets. We have added a sentence reflecting on this in the limitations (lines 263-265).

3. Governments likely to negotiate harder if they own the assets. This conclusion treats government as if it was a private company, which I'm not sure is correct. Governments should be representing all their citizens, including business owners, in negotiations. In many ways it is easier if stranded assets are public - the risk of default is much lower. The people representing the government would not normally personally profit from publicly owned assets?

Thank you for this comment. We agree that governments should not be treated as a private company ignoring its citizens' and businesses' interests at international climate negotiations. Instead we believe that governments with high exposure to stranded assets negotiate harder *because* they care about their citizens and businesses: First, the shut-down of power plants could stress power supply to their citizens and domestic businesses. Second, citizens employed in the power sector as well as upstream and downstream sectors along the value-chain may lose their jobs. Third, businesses dependent on the fossil power sector (including upstream and downstream sectors) are likewise directly or indirectly threatened by stranded assets. Regardless, we have carefully phrased this sentence in the manuscript reflecting that we only hypothesize about the relation between governments' negotiation effort and stranded assets exposure: "These parent owners *may* have high stakes at international negotiations on climate policies" (lines 155-156).

Response to Reviewer #2

Dear colleague,

Thank you very much for your engagement with our manuscript and your recommendation to publish it.

Comments by reviewer #2:

I would to thank the authors for their effort to engage with the reviewers' comments and revise their manuscript.

As far as my comments are concerned, I am satisfied with the treatment of the authors. They provided convincing revisions and replies. I also acknowledge their explanations when some of my suggestions could not be implemented due to lack of data (e.g. weighting lifetime with ROI).

I therefore recommend publication.

Response to Reviewer #3

Dear colleague,

Thank you for your engagement with our manuscript. We include below the entirety of your response and address your concerns point-by-point.

I congratulate the authors to this very effective revision of their initial submission. To me, the stronger focus on resistance by asset owners is compelling, and the paper now a really engaging read.

I have a few minor comments.

Thank you for your kind evaluation. We seek to address your suggestions in what follows.

1. Consider changing your title to reflect the ‘resistance’ aspect. ‘Disentangling’ does not convey a lot of information. You could rather highlight the analysis of potential resistance; and even write something along the lines of “Concentration of asset owners may lead to resistance or some such”. I imagine that this would increase the potential impact your interesting paper has.

Thank you for this comment and the valuable suggestion. We agree and have changed the title to “Concentration of asset owners exposed to power sector stranded assets may trigger climate policy resistance”.

2. In the abstract, you write owners “may face stranded assets of almost 78%”, but looking at your (excellently revised) box plots in figure 7, isn’t that the upper limit? Consider changing slightly to “assets of *up to* 78%”.

Thank you for this suggestion. We have ameliorated the sentence accordingly.

3. In footnote 1, you could also cite Semieniuk et al. (2021) as a thorough discussion of transition risks.

Thank you for this suggestion. We have added the reference in the first paragraph of the discussion (line 254).

4. In the introduction you first only refer to 2degC and then to 1.5 and 2 both on page 1. Can you be consistent one way or the other?

Thank you for this remark. We have changed the degrees consistently to 2°C (line 20).

5. Page 4 bottom, “in contrast to the US” – it is a bit unclear what is being contrasted here. Does the US have a pledge? If it does, what does it not fall short of. I feel an additional sentence setting the stage for this one would help communicate your point.

Thank you for this comment. In response to another reviewer we have changed the sentence so that it does not refer to the US any further (lines 68-70).

6. P. 7, line 105, “than those currently announced” sounds like the stranded assets are announced. Surely you’re referring to policies? Do you mean: “than those under currently announced policies?”

Thank you for this remark and the valuable suggestion, which we have implemented accordingly (lines 106-108).

7. P. 15 l208, I suggest striking ‘ignored’ and instead simply writing “or 80% for those owners where stranded assets do not exceed equity”. (I am not sure how negative equity is relevant to this, as it would result in a negative number).

Thank you for this comment. We have ameliorated the sentence according to your valuable suggestion (lines 211-212).

8. The revised Figure 8 is great. Personally I like showing the actual numbers behind the logs better (the softwares I know allow plotting the original numbers but specifying a ‘log scale’), since it’s not necessarily clear what log 18 really translates to. But at a minimum, the base of the log should be given in the caption.

Thank you for this valuable suggestion. We have altered Figure 8 accordingly (line 226).